# Recognition of Edible Fungi Fruit Body Diseases Based on Improved ShuffleNetV2

**Xingmei Xu, Yuqi Zhang, Hongcheng Cao, Dawei Yang, Lei Zhou and Helong Yu ***

College of Information Technology, Jilin Agricultural University, Changchun 130118, China;
xingmeix@jlau.edu.cn (X.X.); zhangyuqi@mails.jlau.edu.cn (Y.Z.); caohongcheng@mails.jlau.edu.cn (H.C.);
yangdawei@mails.jlau.edu.cn (D.Y.); zhoulei@mails.jlau.edu.cn (L.Z.)
* Correspondence: yuhelong@jlau.edu.cn

**Abstract:** Early recognition of fruit body diseases in edible fungi can effectively improve the quality and yield of edible fungi. This study proposes a method based on improved ShuffleNetV2 for edible fungi fruit body disease recognition. First, the ShuffleNetV2+SE model is constructed by deeply integrating the SE module with the ShuffleNetV2 network to make the network pay more attention to the target area and improve the model's disease classification performance. Second, the network model is optimized and improved. To simplify the convolution operation, the $1 \times 1$ convolution layer after the $3 \times 3$ depth convolution layer is removed, and the ShuffleNetV2-Lite+SE model is established. The experimental results indicate that the accuracy, precision, recall, and Macro-F1 value of the ShuffleNetV2-Lite+SE model on the test set are, respectively, 96.19%, 96.43%, 96.07%, and 96.25%, which are 4.85, 4.89, 3.86, and 5.37 percent higher than those before improvement. Meanwhile, the number of model parameters and the average iteration time are 1.6 MB and 41 s, which is 0.2 MB higher and 4 s lower than that before the improvement, respectively. Compared with the common lightweight convolutional neural networks MobileNetV2, MobileNetV3, DenseNet, and EfficientNet, the proposed model achieves higher recognition accuracy, and its number of model parameters is significantly reduced. In addition, the average iteration time is reduced by 37.88%, 31.67%, 33.87%, and 42.25%, respectively. The ShuffleNetV2-Lite+SE model proposed in this paper has a good balance among performance, number of parameters, and real-time performance. It is suitable for deploying on resource-limited devices such as mobile terminals and helps in realization of real-time and accurate recognition of fruit body diseases of edible fungi.

**Keywords:** edible fungi fruit body; disease recognition; ShuffleNetV2; attention mechanism

## 1. Introduction

China's edible fungi industry is the fifth largest industry after grain, oil, vegetables, and fruits [1,2], which contributes much to the country's economic construction. The development of edible fungi industry provides an important future food source for humans [3,4]. Edible fungi have high protein content, high nutritional value and high medicinal value, and they have significant value as functional foods and for medicinal purposes [5]. With the increasing demand for edible fungi, it is urgent to ensure the production and quality of edible fungi and maintain the healthy development of the edible fungi industry [6–8]. In recent years, the planting scale and types of edible fungi have gradually increased, followed by an increasing number of disease types [9]. There are many edible fungi diseases in the frustum stage, such as brown rot, soft rot, pilum spot disease, brown spot disease, straw mushroom pellet sclerotium disease, etc., which cause serious economic losses [10]. At present, the problems in the prevention and control of edible fungi diseases are the following: there are many types of edible fungi diseases, and mushroom farmers cannot accurately diagnose edible fungi diseases, failing to achieve effective prevention and control in the initial stage of the disease. As a result, disease control is seriously affected, resulting

in a decline in the quality and yield of edible fungi. The fruit body period of edible fungi is a key period for disease prevention and control. Realizing accurate diagnosis of disease types in the fruit body period of edible fungi can provide theoretical guidance for farmers to accurately spray drugs, which is significant for improving the yield of edible fungi [11,12] and ensuring the healthy and sustainable development of the edible fungi industry [13–15].

Traditional machine learning methods, such as naïve Bayes [16–18], logistic regression [19], and support vector machine [20–22], are not suitable for recognizing edible fungi diseases in the fruit body period due to their shortcomings in high computational complexity, slow convergence rate, and difficulty in processing a large number of complex samples [23,24]. In recent years, deep learning methods have been widely studied in crop disease recognition [25–30]. For instance, Nurul Nabilah et al. [31] took 974 pepper disease images collected by themselves and used traditional methods and deep learning methods for experimental comparison. It was found that the recognition accuracy of deep learning was 92.10%, and the deep learning method was significantly better than the traditional feature extraction method. Chen J. et al. [32] proposed an INC-VGGNet model for rice disease recognition based on the VGGNet framework by introducing the Inception module and using the transfer learning method. The accuracy of the model on the public dataset was 91.83%, and the experimental results indicated that the method can be used for disease classification. With the rapid development of deep learning technology, more and more researchers apply lightweight convolutional neural networks to classify and identify crop diseases [33–37]. Chen Junde et al. [38] proposed a Mobile-DANet model that retains the transition layer structure based on DenseNet, replaces the traditional convolutional layer with a depth-wise separable convolutional layer, and introduces an attention mechanism to learn the relationship between channels and the importance of spatial points to input features. Mobile-DANet achieved an average recognition accuracy of 95.86% on locally collected data. Wang et al. [39] proposed an attention-based deep separable Bayesian optimization neural network for recognizing and classifying rice diseases. The model was improved from MobileNet and trained and tested on four categories of rice disease datasets. Atila et al. [40] used the EfficientNet model to identify plant diseases, and after adjusting the hyperparameters, the recognition accuracy was 99.91%. Kang et al. [41] developed an automatic mushroom recognition system using a convolutional neural network. To better investigate the characteristics of mushroom image data, AlexNet, VGGNet, and GoogLeNet were used for comparative experiments, and the class number expansion and fine-adjustment technology were exploited to realize transfer learning. The final top-five accuracy (the probability that the top five categories contain the actual results) was 96.84%. To sum up, although the research on crop disease recognition based on deep learning methods has achieved good results, it still requires much manual intervention in the recognition process, the number of model parameters is large, and the training time is long, which is not conducive to realizing rapid real-time detection of diseases. Therefore, it is necessary to use lightweight convolutional neural networks to design an efficient and non-destructive method for recognizing edible fungi fruit body diseases.

Aiming at the problems mentioned above, this study proposes a lightweight neural-network-based method for recognizing edible fungi diseases in the fruit body period. This paper improves the ShuffleNetV2 network based on the fusion of an attention mechanism and obtains the ShufflenetV2-Lite+SE model [42–47], which is used to recognize edible fungi diseases in the fruit body period. This method provides certain technical guidance for the recognition of edible fungi diseases.

## 2. Materials and Methods

### 2.1. Construction of the Dataset

This study used an image dataset of edible fungi fruit bodies. Part of the data comes from Fusong County, Tonghua County, and other places in Baishan City, Jilin Province. It was filmed by researchers on site, and some of the data comes from the internet and reference books. In order to avoid the problem of misplacing disease categories, we

specifically invited domain experts to screen the data and obtained a dataset containing 649 images of four types (three types of diseases and one type of health), including 224 images of physical disease, 139 images of bacterial disease, 131 images of fungal disease, and 155 images of healthy condition. The images were saved in the jpg format. Some edible fungi fruit body images are shown in Figure 1.

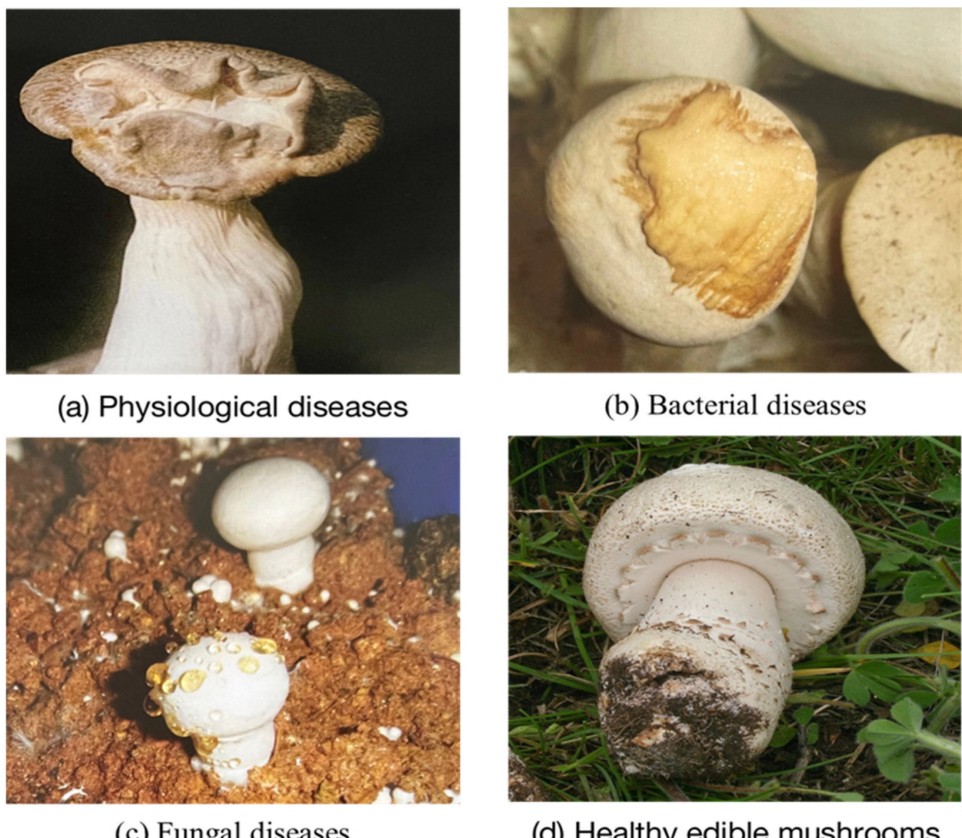

(a) Physiological diseases    (b) Bacterial diseases

(c) Fungal diseases    (d) Healthy edible mushrooms

**Figure 1.** The image types of fruit bodies of edible fungi.

*2.2. Data Preprocessing*

2.2.1. Data Augmentation

The edible fungi fruit body images of three types of diseases and one type of healthy condition in the dataset were enhanced in the same way to increase the difference of training data and improve the generalization ability of the model. The processing operations include adding Gaussian noise, increasing light intensity, decreasing light intensity, vertical flip, and horizontal flip [48,49]. These operations adjust the angle, brightness, and blur of the original image, and the total number of images in the dataset is 3439 after data augmentation. The balance of the test data is ensured by data augmentation [50], the quality and quantity of data samples are improved, and sufficient data samples are provided for training the convolutional neural network [51]. Some examples of data augmentation results are shown in Figure 2.

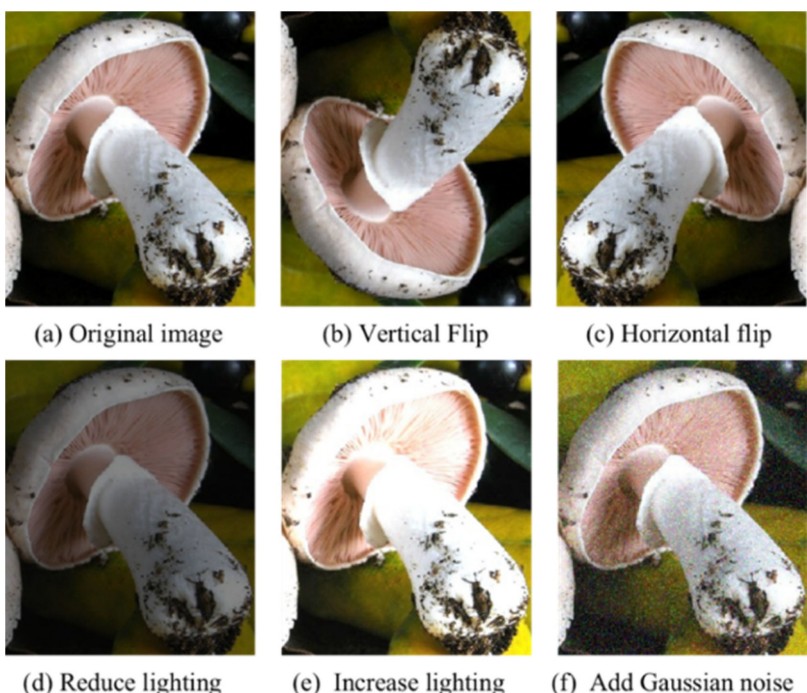

**Figure 2.** Data enhancement results.

2.2.2. Dataset Partitioning

The label of the edible fungi fruit body disease dataset was set with the Python code [52], including four categories: physiological disease, bacterial disease, fungal disease, and healthy condition after data enhancement, which are represented as Physiological, Bacterial, Fungoid, and Health, respectively. Figure 3 shows the statistical chart of the quantity of each type of condition.

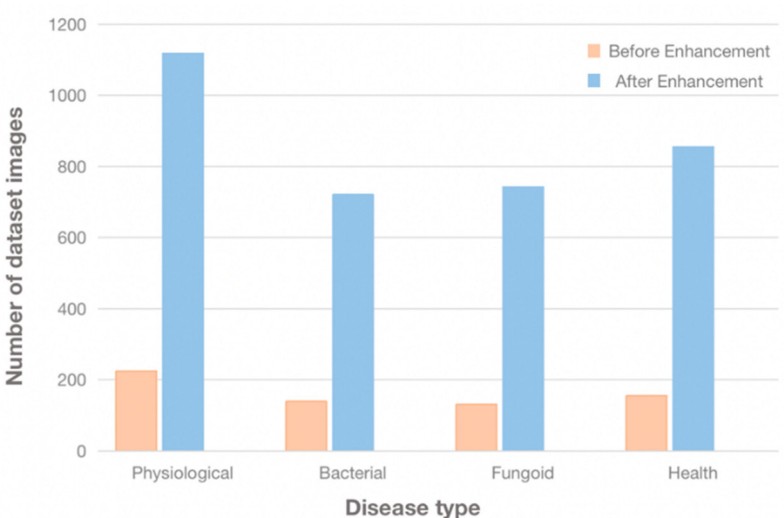

**Figure 3.** The quantity distribution of each type of image.

In this test, the dataset was divided into a training set and a test set at a ratio of 8:2, with 2752 and 687 samples, respectively. The dataset division is shown in Table 1.

**Table 1.** Dataset division.

| Condition | Training Set | Test Set |
|---|---|---|
| Physiological | 895 | 224 |
| Bacterial | 576 | 145 |
| Fungoid | 594 | 149 |
| Health | 684 | 172 |

## 3. Recognition Model of Fruit Body Diseases of Edible Fungi

### 3.1. ShuffleNetV2 Model

ShuffleNetV2 is a lightweight network model proposed by Zhang et al. [53] which aims to greatly reduce the model size and speed up the operation of the model without sacrificing performance efficiency. The biggest innovation of this model is that it fully utilizes the two operations of channel shuffle and group convolution to reduce the calculation amount and the number of parameters of the model. Specifically, channel shuffle is an operation that disrupts the channels of feature maps in order and reconstructs a new feature map to solve the problem of poor information flow caused by group convolution. Excessive group convolution can lead to a large Memory Access Cost (MAC) overhead. In the structure of the ShuffleNetV2 network, the number of output channels of Stage2, Stage3, Stage4, and Conv5 increases successively, while the number of output channels of the stage structure increases successively. As the network depth increases, the feature extraction ability of the model is gradually enhanced, and the detection accuracy is continuously improved. The structure of the ShuffleNetV2 network is shown In Figure 4.

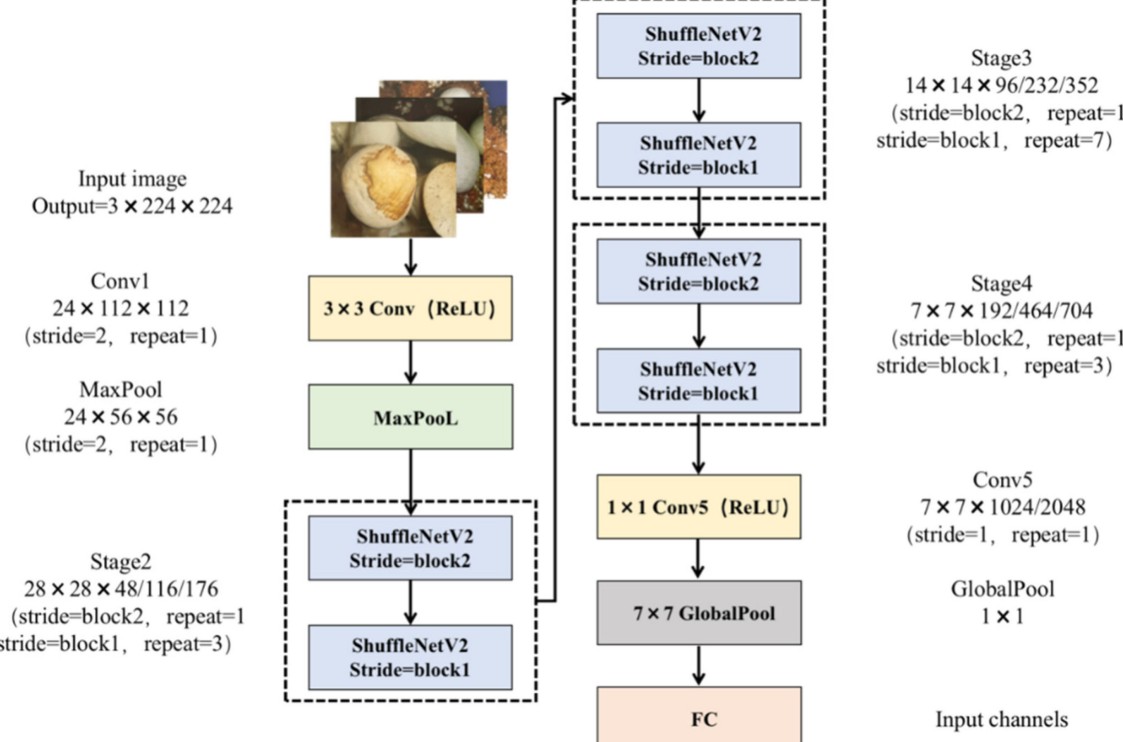

**Figure 4.** The structure of the ShuffleNetV2 network.

The basic structure of the ShuffleNetV2 network consists of two types of blocks. Block1 randomly divides the input channel into two parts: one part retains its mapping and directly transmits downward; the other part performs separable convolution to extract image features. The output channel of the two parts is combined at the bottom of the module. Then, a random mixing operation is performed on the final output feature graph

channel, and the structure is shown in Figure 5a. Block2 sends all feature diagrams into two network branches, and the output feature diagrams of the two branches are combined at the bottom of the module to double the number of final output channels.

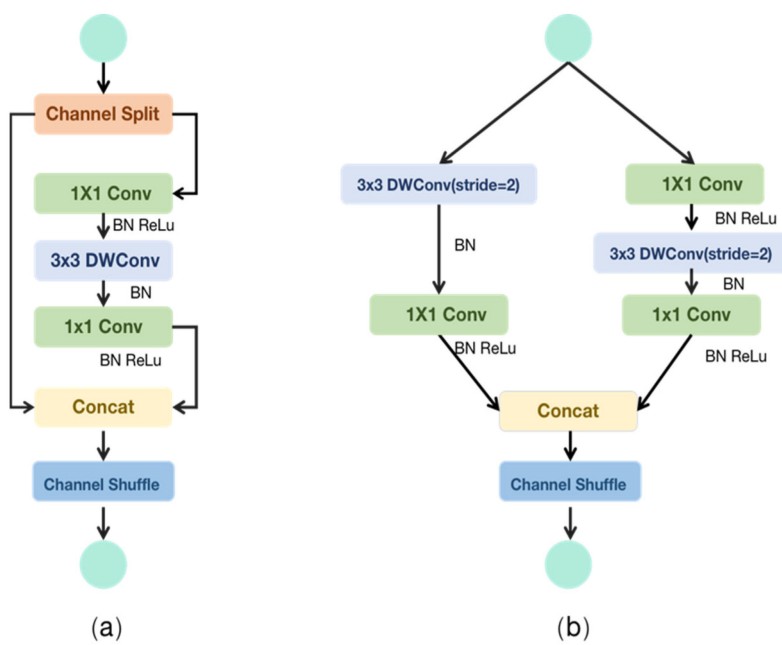

**Figure 5.** (**a**) Basic unit. (**b**) Spatial down sampling unit (2×).

The structure is shown in Figure 5b.

In ShuffleNetV2, all feature channels have the same weight, and the number of channels doubles every time Block2 passes through. With the doubling of the number of channels, much attention is paid to the feature channels that have a great impact on the classification results. Meanwhile, the depth-separable convolution used in Block2 is sensitive to the location of sensitive features, and too much background information is retained, which easily affects the classification effect.

### 3.2. Model Improvement

There are some problems with the disease dataset of edible fungi fruit body constructed in this study, such as complex disease image background, high similarity between some diseases and background, large difference in disease size, etc. As a result, the overall recognition is difficult, and the existing model has poor recognition performance. Considering the recognition accuracy and speed of the model, the ShuffleNetV2 model is improved in this study.

#### 3.2.1. Attention Mechanism

The attention mechanism aims to focus on areas of interest and try to suppress the role of areas of interest in image segmentation as much as possible. In deep learning CNN, attention mechanisms can be divided into two types: channel attention and spatial attention. The channel attention refers to determining the weight relationship between different channels, enhancing the weight of key channels, and suppressing channels with little inhibitory effect. The spatial attention is the determination of the weight relationship between different pixels in the spatial neighborhood, enhancing the weight of key area pixels, allowing the algorithm to pay more attention to the research area we need, and reducing the weight of unnecessary areas.

The SE (Squeeze-and-Excitation) attention mechanism is a method of determining weights in channel attention mode, which achieves priority by assigning weights between different channels. The SE attention module can adjust the weight according to different

feature channels, automatically enhance the feature channels with rich contrast information in the image, and effectively suppress the feature channels unrelated to the target. The model structure is shown in Figure 6a.

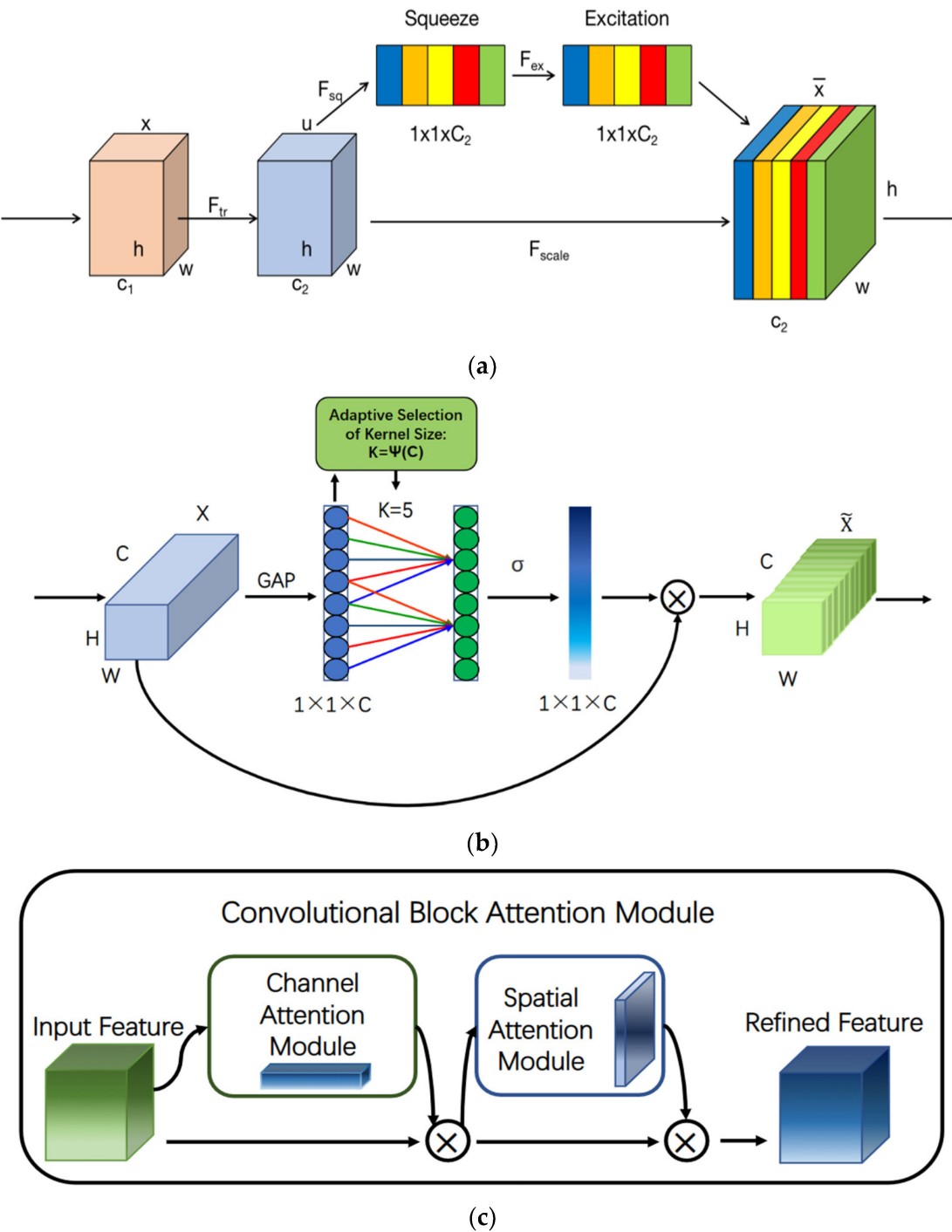

**Figure 6.** (**a**) The structure of the SE module. (**b**) The structure of the ECA module. (**c**) The structure of the CBAM module.

The ECA (Efficient Channel Attention) attention mechanism is a method of channel attention. To avoid damaging the direct correspondence between channels and their weights, a one-dimensional convolution method is proposed to avoid the impact of dimensionality reduction on data. The model structure is shown in Figure 6b.

The CBAM (Convolutional Block Attention Module) attention mechanism is an algorithm model that combines channel and spatial attention mechanisms. The input feature map is first subjected to channel attention mechanism, followed by spatial attention mechanism operation, and, finally, output, achieving the goal of strengthening the region of interest from both channel and spatial aspects. The model structure is shown in Figure 6c.

### 3.2.2. Simplify Model Structure

Among the basic components of ShuffleNetV2, three convolution layer operations go through on the right branch, which are the normal $1 \times 1$ convolution layer (performing Batch Normalization, BN, and ReLU), the $3 \times 3$ deep convolution layer (performing BN), and the normal $1 \times 1$ convolution layer (performing BN ReLU). Two $1 \times 1$ convolution layer operations are used here, but there are actually some superfluous layers. Because the dimensionality-up and dimensionality-down operation is not necessary here, only the fusion of the inter-channel information of the DW convolution using a $1 \times 1$ convolution layer operation is sufficient. Therefore, in this study, deletion of a $1 \times 1$ convolution layer after a $3 \times 3$ deep convolution layer is considered to achieve the goal of model lightweight. The improved ShuffleNetV2-Lite network structure is shown in Figure 7.

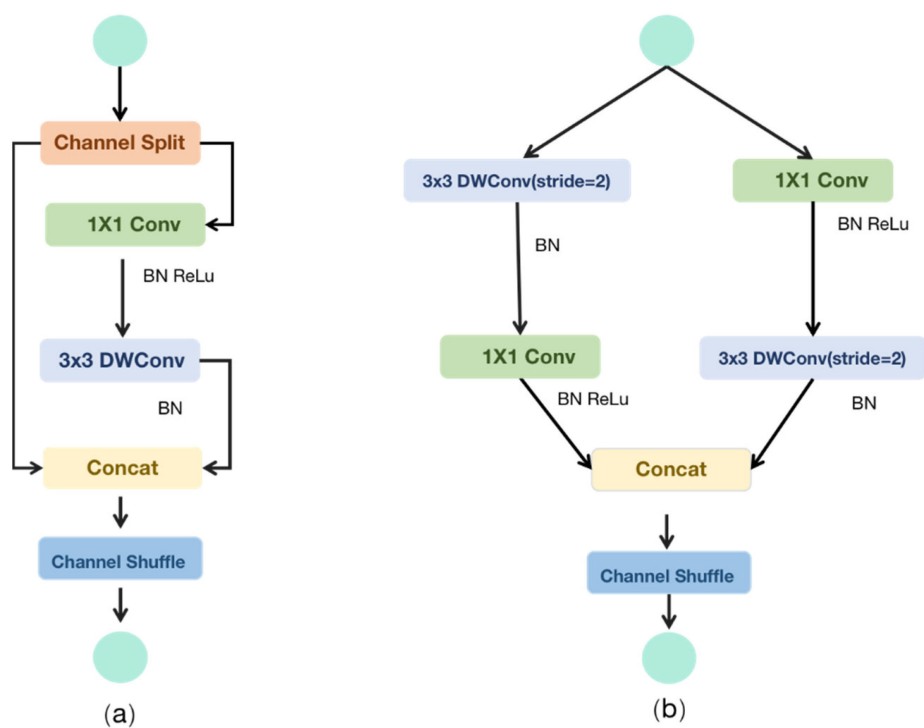

**Figure 7.** (**a**) The improved basic unit. (**b**) The improved Spatial down from the unit (2×).

In this study, the redesigned ShuffleNetV2-Lite network structure can effectively reduce computational complexity and optimize model performance while ensuring model accuracy.

### 3.3. Experimental Environment and Hyperparameters

Python3.8 and CUDA11.3 are configured under the Windows 10 operating system to build a convolutional neural network structure with the Pytorch framework as the core. The parameters of the experimental environment are shown in Table 2.

**Table 2.** The parameters of the experimental environment.

| Environment | Parameters |
|---|---|
| OS | Windows 10 |
| CPU | Intel Xeon Gold 6246R @ 3.40 GHz 32 cores |
| Memory | 128 GB |
| Deep learning framework | Pytorch-GPU 1.12.1 |
| GPU | NVIDIA Quadro RTX 8000 |
| CUDA version | CUDA Toolkit 11.3 |
| Pytorch version | Pytorch 0.9.1 |

The parameters for model training are shown in Table 3: the input image size is 224 × 224 pixels, the number of training iterations is 200, the batch size is 64, the learning rate is 0.01, the optimizer is Adam, and the cross-entropy loss function is adopted. To prevent overfitting and under fitting, L2 regularization method and the Class_Weighting loss function weighting method are added to the cross-entropy loss function of the model. We increased the weight of the loss function for the small class sample data of the classifier and reduce the weight of the loss function for the large class sample to solve the problem of data imbalance.

**Table 3.** Parameters for model training.

| Parameters | Values |
|---|---|
| Epoch | 200 |
| Batch size | 64 |
| Learning rate | 0.01 |
| Optimizer | Adam |

*3.4. Evaluation Index*

This study uses four indicators, accuracy, precision, recall, and Macro-F1, to measure the model performance.

(1)  Accuracy represents the proportion of correct results predicted by the model in the total samples. The calculation formula is

$$\text{Accuracy} = \frac{\text{TP} + \text{TN}}{\text{TP} + \text{TN} + \text{FP} + \text{FN}}. \tag{1}$$

(2)  Precision represents the proportion of positive samples predicted correctly by the model in the predicted positive samples. The calculation formula is

$$\text{Precision} = \frac{\text{TP}}{\text{TP} + \text{FP}}. \tag{2}$$

(3)  Recall represents the proportion of positive samples predicted by the model to the actual positive samples. The calculation formula is

$$\text{Recall} = \frac{\text{TP}}{\text{TP} + \text{FN}}. \tag{3}$$

(4)  Macro-F1 represents the harmonic mean of precision and recall, and it reflects the comprehensive performance of the model. The calculation formula is

$$\text{Macro-F1} = 2 \times \frac{\text{Precision} \times \text{Recall}}{\text{precision} + \text{Recall}}, \tag{4}$$

where TP indicates that the predicted positive sample is also the actual positive sample; FP indicates that the predicted positive sample is actually negative; FN indicates that the

predicted negative sample is actually positive; TN indicates that the predicted negative sample is also actually negative.

## 4. Experimental Results and Analysis

### 4.1. Model Performance Evaluation with Different Attention Mechanisms

In this experiment, the Convolutional Block Attention Module (CBAM) module, the Efficient Channel Attention (ECA) module, and the Squeeze-and-Excitation (SE) module were embedded into ShuffleNetV2, respectively, and the experiments were conducted on the edible fungi fruit body disease image dataset. The performance comparison of different attention mechanisms is shown in Table 4.

**Table 4.** Performance comparison of different attention mechanisms.

| Model | Accuracy (%) | Params (MB) | Time/Epoch (s) |
|---|---|---|---|
| ShuffleNetV2 | 91.34 | 1.4 | 45 |
| ShuffleNetV2+CBAM | 92.91 | 1.9 | 57 |
| ShuffleNetV2+ECA | 93.15 | 2.3 | 61 |
| ShuffleNetV2+SE | 93.77 | 1.8 | 47 |

The recognition accuracy of the ShuffleNetV2+SE model was 93.77%, which was 2.43%, 0.14%, and 0.62% higher than that of ShuffleNetV2, ShuffleNetV2+CBAM, and ShuffleNetV2+ECA models, respectively. It can be seen that by adding the attention mechanism, the information interaction between channels can be better realized, and the performance of the model can be improved. Especially after adding the SE module, the model pays more attention to the channel features with the most information, and unimportant channel features are suppressed. In this experiment, more attention was paid to the disease area of the edible fungi fruit body, higher disease recognition accuracy was obtained, and the model achieved the best performance.

The parameter of the ShuffleNetV2+SE model was 1.8 MB, which is 0.4 MB more than that of the original model. The parameters of the ShuffleNetV2+CBAM model and the ShuffleNetV2+ECA model were reduced by 0.1 MB and 0.5 MB, respectively. The average iteration time of the ShuffleNetV2+SE model was 47 s, showing an increase of 2 s compared with the original model. Compared with the ShuffleNetV2+CBAM model and the ShuffleNetV2+ECA model, the average iteration time was reduced by 10 s and 14 s, respectively. In terms of model parameters, the SE module has the best performance among the three attention mechanisms. In terms of training time, adding the attention mechanism can effectively reduce the average iteration time. Considering model recognition accuracy, the number of model parameters, as well as the average iteration time, the SE module was chosen to be embedded in ShuffleNetV2 to construct the model in this study.

### 4.2. Ablation Experiments for the ShuffleNetV2-Lite+SE Model

The attention SE module is introduced into the ShuffleNetV2 model, and the structure of the model is optimized to obtain the ShuffleNetV2+SE model, the optimized ShuffleNet-Lite model, and the ShufflenetV2-Lite+SE model, respectively. To investigate the performance improvement effect of the ShuffleNetV2 model brought by the attention mechanism and the structural optimization, ablation experiments were conducted. The performance of the model was evaluated in terms of accuracy, precision, recall, and Macro-F1 value on the test set, and the number of model parameters and the average iteration time were used to evaluate the complexity of the model. The performance comparison of different models is shown in Table 5.

**Table 5.** Performance comparison of different models.

| Model | Accuracy (%) | Precision (%) | Recall (%) | Macro-F1 (%) | Params (MB) | Time/Epoch (s) |
|---|---|---|---|---|---|---|
| ShuffleNetV2 | 91.34 | 91.54 | 92.21 | 90.88 | 1.4 | 45 |
| ShuffleNetV2+SE | 93.77 | 93.84 | 93.59 | 93.71 | 1.8 | 47 |
| ShuffleNetV2-Lite | 93.88 | 93.17 | 94.70 | 93.93 | 1.2 | 40 |
| ShuffleNetV2-Lite+SE | 96.19 | 96.43 | 96.07 | 96.25 | 1.6 | 41 |

As shown in Table 5, the accuracy, precision, recall, and the Macro-F1 value of the ShuffleNetV2-Lite+SE model reach 96.19%, 96.43%, 96.07%, and 96.24%, respectively. The accuracy, precision, recall, and the Macro-F1 value are improved by 2.42, 2.59, 2.48, and 2.54 percent, respectively. Compared with other models, the proposed model achieves higher accuracy and better performance.

The parameters and the average iteration time of the ShufflenetV2-Lite+SE model are 1.6 MB and 41 s, respectively. Compared with the ShuffleNetV2+SE model with the highest accuracy, the model parameters are reduced by 11.11 percent, and the iteration time is reduced by 12.77 percent. Thus, the proposed model has lower model complexity than other models.

The improved model proposed in this study shows a better balance between the performance, complexity, and real-time performance of the model. It is suitable for deploying on embedded resource-constrained devices such as mobile terminals and helps to realize real-time and accurate recognition of edible fungi fruit body diseases.

### 4.3. Performance Comparison of ShuffleNetV2-Lite+SE with Other Models

To further evaluate the recognition effect of the improved network model on edible fungi disease images, the ShuffleNetV2-Lite+SE model was compared with the representative lightweight convolutional neural networks including MobileNetV2, MobileNetV3, DenseNet, and EfficientNet on the same dataset under the same experimental environment and network parameter configuration. The accuracy, precision, recall, and Macro-F1 value of these models on the test set were used to evaluate the model performance, and the number of model parameters and the average iteration time were used to evaluate the complexity of the model. The performance comparison of ShuffleNetV2-Lite+SE with other models is shown in Table 6, and the accuracy of the model on the test set, the change in loss value, and the confusion matrix during iterations are shown in Figures 8–10.

**Table 6.** The performance comparison of ShuffleNetV2-Lite+SE with other models.

| Model | Accuracy (%) | Precision (%) | Recall (%) | Macro-F1 (%) | Params (MB) | Time/Epoch (s) |
|---|---|---|---|---|---|---|
| MobileNetV2 | 85.72 | 85.33 | 85.93 | 85.63 | 3.5 | 66 |
| MobileNetV3 | 91.72 | 91.38 | 91.40 | 91.39 | 5.1 | 60 |
| DenseNet | 88.50 | 88.58 | 87.04 | 87.80 | 7.8 | 62 |
| EfficientNet | 89.29 | 89.46 | 89.98 | 89.84 | 5.3 | 71 |
| ShuffleNetV2-Lite+SE | 96.19 | 96.43 | 96.07 | 96.25 | 1.6 | 41 |

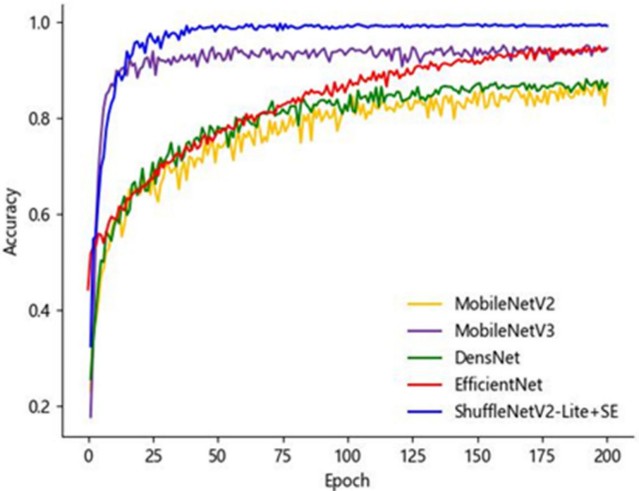

**Figure 8.** The accuracy of different models.

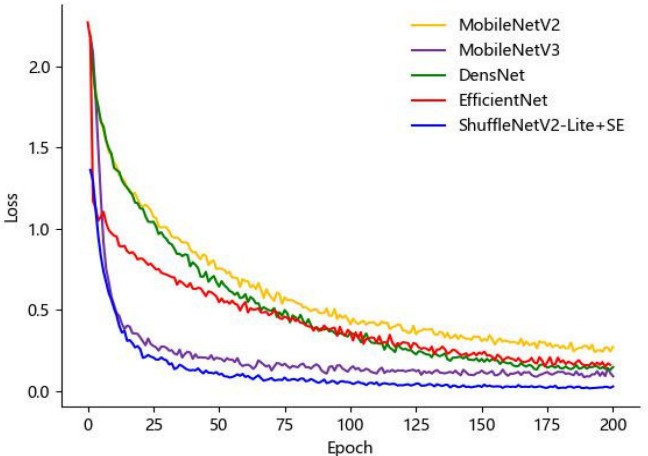

**Figure 9.** The loss values of different models.

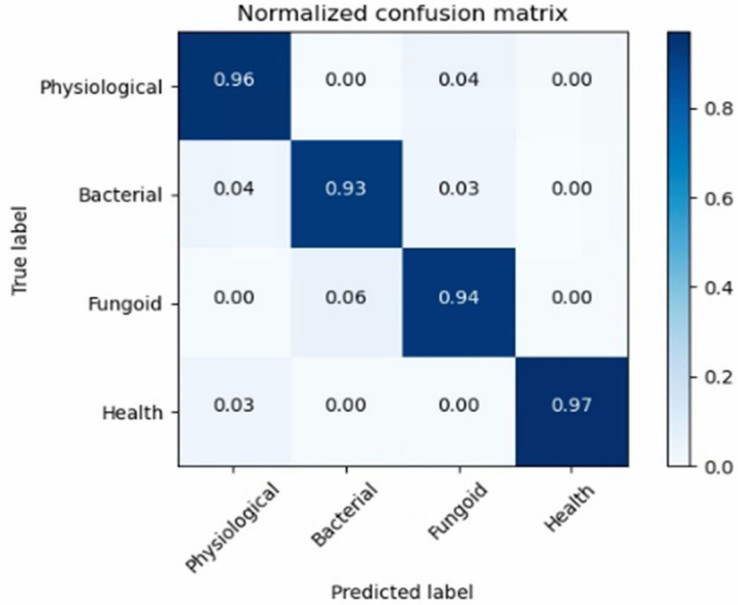

**Figure 10.** The confusion matrix.

It can be seen from Table 6 that the ShuffleNetV2-Lite+SE model achieves an accuracy of 96.19%, a precision of 96.43%, a recall of 96.07%, and a Macro-F1 value of 96.24%, respectively. Compared with the best-performing MobileNetV3 model, the accuracy, precision, recall, and Macro-F1 value are improved by 4.47, 5.05, 4.67, and 4.86 percent, respectively. Compared with other models, the proposed model has higher accuracy and better performance.

The parameters and the average iteration time of the ShuffleNetV2-Lite+SE model are 1.6 MB and 41 s, respectively. Compared with the best-performing MobileNetV3 model, the parameters of the ShufflenetV2-Lite+SE model are reduced by 68.63%, and the average iteration time is reduced by 31.67%, indicating that the ShufflenetV2-Lite+SE model has lower model complexity than other models.

It can be seen from Figure 8 that the accuracy of the five models gradually flattens out as the number of iteration rounds increases. After the 20th iteration round, the accuracy curve of the ShuffleNetV2-Lite+SE model is relatively smooth, there is no over-fitting phenomenon, and the accuracy becomes stable at more than 90%, while the accuracy of the other models only reaches about 80%. After the 70th iteration round, the accuracy of all models tends to be stable, and the accuracy of the ShuffleNetV2-Lite+SE model reaches 96.19%, which is the highest among all models. The results indicate that the network has good recognition accuracy.

Figure 9 shows that in the initial training stage, the loss values of all networks continuously decrease and eventually tend to be stable without large fluctuations. The ShuffleNetV2-Lite+SE model has a significantly better convergence speed than other models. After the 29th iteration round, the loss value of the ShuffleNetV2-Lite+SE model remains below 0.2, and after the 55th iteration round, the overall loss value curve change becomes stable, and the loss value of the five models is lower than 0.5. In addition, the loss value of the ShuffleNetV2-Lite+SE model is significantly lower than that of the other models, showing a higher classification ability of this model.

Figure 10 shows the classification confusion matrix of the ShuffleNetV2-Lite+SE model for images of the three types of diseases and one type of health. It can be seen that the prediction accuracy of the images of the physiological disease is 96%, the prediction accuracy of the images of the bacterial disease is 93%, and the prediction accuracy of the images of the fungoid disease is 94%. The prediction accuracy of the images of the healthy condition is 97%. Among them, the accuracy of disease image prediction is relatively low for the bactrial and fungoid groups. The reason is that there are problems in the dataset of edible fungi fruit body diseases, such as complex disease image backgrounds, high similarity between some diseases and the background, and significant differences in disease sizes, which cause the model to misjudge. Overall, the ShuffleNetV2-Lite+SE model has a good recognition effect on the four types of images.

## 5. Conclusions

In this study, the recognition of fruit body diseases of edible fungi was taken as the research object, and the problems of low recognition accuracy, complex model structure, and slow recognition speed of the existing model were addressed based on the improvements of ShuffleNetV2.

(1) The constructed edible fungi fruit body disease dataset contains images of three types of diseases and one type of health condition, with a total of 649 images. After data augmentation, the total number of images in the dataset is 3439.

(2) The CBAM module, ECA module, and SE module are embedded into ShuffleNetV2, respectively, to enhance the information interaction between channels and improve the model's performance. Experimental results indicate that the recognition accuracy of the ShuffleNetV2+SE model on the test set is 2.43 percent higher than that of the original model. Compared with the ShuffleNetV2+CBAM model and the ShuffleNetV2+ECA model, the recognition accuracy of the Shufflenetv2+SE model is improved by 2.43 percent; the number of model parameters of the ShuffleNetV2+SE

model is reduced by 0.1 MB and 0.5 MB, and the average iteration time is reduced by 10 s and 14 s, respectively.

(3) The attention SE module is introduced into the ShuffleNetV2 model, and the structure of the model is optimized to obtain the ShuffleNetV2+SE model, the optimized ShuffleNet-Lite model, and the ShufflenetV2-Lite+SE model, respectively. Experimental results indicate that the accuracy, precision, recall, and the Macro-F1 value of the ShuffleNetV2-Lite+SE model reach 96.19%, 96.43%, 96.07%, and 96.24%, respectively, which is higher than those of other models. Compared with the ShuffleNetV2+SE model with the highest accuracy, our model reduces the number of model parameters by 11.11%, and the average iteration time by 12.77%, so it has lower model complexity than other models.

(4) The ShuffleNetV2-Lite+SE model is compared with representative lightweight convolutional neural networks, including MobileNetV2, MobileNetV3, DenseNet, and EfficientNet. Compared with the best-performing MobileNetV3 model, the parameters of the ShufflenetV2-Lite+SE model are reduced by 68.63%, and the average iteration time is reduced by 31.67%. The experimental results show that the ShuffleNetV2 Lite+SE model has higher accuracy and lower model complexity, and has certain advantages compared to the existing relevant research. It can be deployed on mobile terminal devices to promote real-time and accurate recognition of diseases in edible fungi fruit bodies.

The ShuffleNetV2-Lite+SE model proposed in this study reduces the model complexity and considers the recognition accuracy and speed of edible fungi fruit body diseases, which lays the foundation for the recognition of edible fungi fruit body diseases. The next step will be to integrate the outdoor environment and factory planting image data and attempt to transplant the model to mobile platforms to test its real-time recognition effect on edible fungi fruit body diseases, providing assistance for the prevention and control of edible fungi fruit body diseases.

**Author Contributions:** The contributors are X.X. and Y.Z. for conceptualization; H.C. for methodology; investigation, D.Y. and L.Z.; H.Y. for formal analysis; X.X. and Y.Z. for investigation/writing—original draft/supervision; X.X. and H.Y. for visualization; H.Y. for writing—review/editing. All authors have read and agreed to the published version of the manuscript.

**Funding:** This work was supported by the Natural Science Foundation of Jilin Province (YDZJ202201ZYTS544) and the Technology Development Plan Project of Jilin Province (20200403176SF).

**Data Availability Statement:** Data supporting the findings of this study are available from the corresponding author.

**Conflicts of Interest:** The authors declare no conflict of interest.

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
