# Peer review of "Recognition of Edible Fungi Fruit Body Diseases Based on Improved ShuffleNetV2"

_agronomy, doi:10.3390/agronomy13061530_

Round 1
Reviewer 1 Report
It would be nice to test the model presented in the paper also on other datasets.
I would mention on contributions the dataset creation.
Is this dataset and the source code available online? If yes, could you add a link to them? Thanks
Reviewer 2 Report
Summary
The research described in the manuscript problematizes the identification of diseases of the fruiting body of edible fungi. The fruiting period of edible body fungi is considered a key period for disease prevention and control. Therefore, early identification of these diseases can effectively improve the quality and yield of edible fungi, which is a very important demand. In their arguments, the authors highlight the justification for carrying out the present research, emphasizing that traditional machine learning methods have limitations for the recognition of diseases of the fungal fruiting body due to their deficiencies in high computational complexity, low convergence rate and difficulty in processing a large number of complex samples. In this sense, an improved method based on ShuffleNetV2 is proposed for the recognition of diseases of fruiting bodies of edible fungi with the integration of the SE module in order to improve the performance of the classification of diseases, resulting in ShuffleNetV2+SE that was later optimized and improved. The materials and methods are well described and concise, with enough detail that they can be replicated. The main question posed in the research was well addressed and in the results the authors presented their main findings well. The proposed model ShuffleNetV2-Lite+SE showed to be promising, obtaining better performance when compared to usual models. In addition, the method proved to be suitable for use in devices with limited resources and can help, in a practical way, in the accurate identification of fungal edible fruit diseases. That's actually a big contribution.
General concept comments
The topic is relevant and the text is generally well articulated. The materials and methods section presents important information. However, the models used for comparison with the proposed method were not first described in this section. They only appear in results. It would be important to put this information in “materials and methods” to make it clearer to the reader what will be done in the work. In addition, in the discussion of the results, studies in the literature are not cited. Papers could be discussed to give a perspective on the use of the proposed model and in what these models could add to the resolution of the general problem addressed. The conclusions could also be stated more concisely and directly. Several pieces of information are introduced before the conclusions that have already been stated, for example in the introduction or methodology.
Revision
The research topic is relevant and the authors highlight the justification for their research. The limitations of traditional methods are well emphasized in the introduction. As well as the importance of developing more efficient and practical tools that help in the identification of diseases in a timely manner to enable the demand for yield and quality of edible fungi. Updated references were highlighted during the introduction, allowing the reader to better understand the state of the art of the subject, limitations of usual methods, justifications and perspectives of the proposed research.
The experimental design and control methods used proved to be appropriate in conducting the study. The materials and data fitting used were well detailed. The information is clear and allows the replication of research by specialists. However, I believe, in terms of organization, that the models used for comparison with the proposed method should be present in the materials and methods section.
The results focused on the main findings and also sought to highlight the objectives set for this research. Tables and graphs were arranged in appropriate quantities and qualities to illustrate the comparison of methods and other interpretations. I only emphasize that the titles of the figures should be self-explanatory to improve the reading and interpretation of the results to any reader. One point I would like to highlight is that the results did not mention references for discussion purposes. That is, to highlight some works carried out with the same theme and possible discussions, showing perspectives in relation to the results of the present research.
In the “conclusions” subsection, the authors conscientiously highlighted their main results: according to the objectives, evidence and arguments presented throughout the text. However, the text could be improved by focusing only on the main findings directly. The text could be improved if it were concisely rephrased, drawing on information that was already provided in the introduction or materials and methods.
The references used by the authors are relevant, current and appropriate, especially in the sections: “Introduction” and “Materials and method”. As already emphasized, other references, or even the same ones already cited, could be used in “Results” for discussions further enriching the work.
Specific Comments
In general, the organization and text of the manuscript are of good quality. However, I would like to highlight some points that could be improved and contribute to make the text stronger.
1) The information present in the first paragraph of the introduction (lines 357 to 360) should be in the “materials and methods” section. That is, mentioning all the models used for comparison in the “Materials and methods” section would be important to make it clear to the reader what will be analyzed in the research.
2) I recommend that authors discuss their findings, in Results, citing some work, to give some perspectives, for example, what their findings could advance in relation to the cited study, etc.
3) In the conclusions, the authors present a contextualization before presenting the actual conclusions. This information has already been opportunely placed in Introduction or Materials and Methods. I believe that in “conclusions” the text should be more concise and direct.

Reviewer 3 Report
The paper proposed an interesting result on the recognition of edible fungi disease. The architecture in use is some modifications to the existing architecture ShuffleNet V2. There are several points that require some clarifications:
- line 18: "The number of model parameters and each ..." I believe the author means the average iteration time. Reduction from average iteration time should be easy since one layer is removed, but without the standard deviation of the changes, it is not a valid statistical comparison.
- It is nice to have a comparison of the result with DeepLabV3+ as the state-of-the-art method. This will allow us to check the performance not only within lightweight models.
- line 133: "The balance of the test data....", what about the balance in the training data. Figure 3 shows that the training data is imbalanced. How do you check whether there is or not a catastrophic forgetting while training the model.
- Figure 1 shows there are different fungi species are considered in the study. In the previous question, for each species, we need to have the same number of elements as other species in each class. Also, it is required to check whether there are similar diseases or possible anomalies that can be incorrectly classified as healthy or diseases.
- line 294: "Two ordinary 1×1 convolutions are slightly redundant". What does slightly redundant means?
- Line 336, Table 3: As stated after augmentation 3439 data points within 4 classes are used. And 80 % of these data are used for training (around 2750 in total). The model is trained for 200 epochs and batch size 64 this means that 12800 images are used in total for training. This is clearly overfitting.
- In Figure 10 If we look at the main diagonal, it shows the effect of imbalanced training data. There might be some catastrophic forgetting in the training.
The English can be further improved for example:
in line 8: body diseases of edible fungi - > body diseases in edible fungi
in lines 11 and 13 firstly and secondly - > first and second
in line 25 parameter quantity - > number of parameters
in line 26 resource-constrained - > resource-limited
This is just in abstract.
in line 27: helps to realize - > helps in
Reviewer 4 Report
The research entitled (Based on the improved ShuffleNetV2 edible fungus fruiting body disease recognition) and entitled (Recognition of Edible Fungi Fruit Body Diseases Based on Improved ShuffleNetV2) in the website. Proposes a lightweight neural network-based method for recognizing edible fungus fruiting body disease. The research is not suitable for agronomy journal for many reasons.
1. First the authors didn’t know what are mushroom types included in this study they only mentioned them as edible mushrooms.
2. There isn’t any information about the mushroom in the photos, sources, conditions, …. Etc.
3. The research quality is very low, with low figure resolutions, and low validations of the methods.
4. The title of the manuscript on the web site not the same as the manuscript.
5. The authors did not follow the journal reference style.
6. The research needs extensive English editing.
The research needs extensive English editing
Round 2
Reviewer 3 Report
Most of the comments are not answered:
- line 18: "The number of model parameters and each ..." I believe the author means the average iteration time. This is not answered.
- Imbalance of training data: the author's response to the comment is by addressing Figure 3 which clearly shows the data imbalance.
- Convolution slightly redundant: the author's response to the comment shows the author means: "Experimental comparison shows that the use of two ordinary 1x1 convolutions superfluous". It has nothing to do with being redundant. The term "redundant" has other meanings.
- Problem of overfitting: the author's response to the comment was that the experimental results show that there is no overfitting, but the parameters used are textbook overfitting, also I could not find any early stopping there. And they suggested an increase of data to prevent overfitting, but this is not true most of the time for example assume data points with observed linear patterns but fit with a polynomial of degree say 3. It does not matter how many additional data points will be added the polynomial fit is an overfitting.
- Catastrophic forgetting: Figure 10 shows the effect of imbalanced data and there is a possibility of catastrophic forgetting. The authors' response was only explaining the figure.
The English language is only modified locally, the authors need to go through the paper and make an effort to improve the paper. For example:
line 32: ...., and fruit -> ...., and fruits
line 33: development of the edible -> development of edible
lines 34-35: "Edible fungi contain high protein and have high nutritional value and medicinal value, and it has significant value as a functional food and in medicinal use" -> Edible fungi have high protein content, high nutritional value and high medicinal value, and they have significant value as functional foods and for medicinal purposes
line 41: which will cause -> which cause
Reviewer 4 Report
The research has many defects in the design, quality, and missing significant informations.
Moderate revision
Round 3
Reviewer 3 Report
In response to Imbalance data, the author's response was the use of the L2 regularization method. While the L2 regularization helps to prevent overfitting by penalizing the loss function, it does not address the problem of imbalance. As we know L2 regularization also known as weight decay does not explicitly target class samples but instead focuses on penalizing large weights in the model. It also does not reduce the weight of the loss function for specific class samples. However, the statement has some valid points but it does not answer the problem. Because L2 regularization does not adjust the weights of the cross-entropy loss function on class sample sizes, so the main problem still presist.
Reviewer 4 Report
The authors did a good work of enhancing the manuscript. It could be accepted after revising the reference list one by one and adding the missing page numbers.
Round 4
Reviewer 3 Report
Making code available in Github might be good for more use cases. The explanations are correct now. However, I cannot understand why the results in the main body of the paper give the impression that data imbalance is not considered in the use case scenario.